# Effects of Mindfulness-Based Stress Reduction Training on Healthcare Professionals’ Mental Health: Results from a Pilot Study Testing Its Predictive Validity in a Specialized Hospital Setting

**DOI:** 10.3390/ijerph17249420

**Published:** 2020-12-16

**Authors:** Math Janssen, Beatrice Van der Heijden, Josephine Engels, Hubert Korzilius, Pascale Peters, Yvonne Heerkens

**Affiliations:** 1Occupation & Health Research Group, HAN University of Applied Sciences, 6525EN Nijmegen, The Netherlands; josephine.engels@han.nl (J.E.); yvonne.heerkens@han.nl (Y.H.); 2Institute for Management Research, Radboud University, 6525AJ Nijmegen, The Netherlands; b.vanderheijden@fm.ru.nl (B.V.d.H.); h.korzilius@fm.ru.nl (H.K.); p.peters@nyenrode.nl (P.P.); 3The Netherlands School of Management, Open University of the Netherlands, 6419AT Heerlen, The Netherlands; 4The Netherlands Faculty of Economics and Business Administration, Ghent University, 9000 Ghent, Belgium; 5Hubei Business School, Hubei University, Wuhan 368 Youyi Ave., Wuchang District, Wuhan 430062, China; 6Kingston Business School, Kingston University, London KT11LQ, UK; 7Center for Strategy, Organization and Leadership, Nyenrode Business Universiteit, P.O. Box 130, 3620AC Breukelen, The Netherlands

**Keywords:** mindfulness, mindfulness-based stress reduction (MBSR), mental health variables

## Abstract

This pilot study aimed to evaluate the feasibility and acceptability of a Mindfulness-Based Stress Reduction (MBSR) training and to examine positive and negative symptom-focused mental health variables. The mental health variables were used to test the predictive validity of the training among healthcare professionals. Thirty healthcare professionals participated in this non-randomized pre-post intervention pilot study. The questionnaire on mental health was filled in twice. Baseline and post-intervention differences were tested with paired samples *t*-tests and Wilcoxon signed-rank tests. The participants’ evaluation of the training was assessed with a five-item questionnaire. The recruitment and retention were successful, and participants’ evaluation of the training itself was positive but the influence on daily life was rated only moderately positive. In comparison with baseline at post-intervention participants showed significant improvements in general mindfulness, the burnout dimension personal accomplishment, quality of sleep, positive emotions, and self-efficacy. A significant decrease was found in the burnout dimension emotional exhaustion, stress level, negative emotions at work, and worrying. No significant changes were found for the burnout dimension mental distance, and work engagement. The measures showed ample within-person differences and low, medium, or high effect sizes. The current trial approach of the MBSR training seems feasible and acceptable. Our results suggest that mindfulness, burnout, stress level, quality of sleep, positive emotions at work, negative emotions at work, self-efficacy, and worrying are meaningful mental health variables for inclusion in a larger-scale Randomized Controlled Trial on the effects of MBSR.

## 1. Introduction

Work-related stress is an inherent feature of industrialized nations [1]. The World Health Organization (WHO) [1] p. 13 defines work-related stress as “a pattern of reactions that occurs when workers are presented with work demands not matched to their knowledge, skills or abilities and which challenge their ability to cope.” According to the European Agency for Safety and Health at Work [2], 51% of the workers in Europe consider work-related stress to be common in their workplace. In the Netherlands, 59% of workers believe that work-related stress is prevalent [2]. In European workplaces, the most commonly perceived causes of work-related stress are job reorganization or job insecurity (72%) and workload or hours worked (66%) [2]. Empirical research has found that work-related stress has effects like decreased productivity at work and increased absenteeism and work-related turnover [3,4].

Working in healthcare is particularly stressful, which is reflected in the fact that 61% of European healthcare professionals experience work-related stress [2]. In comparison with other categories of workers, healthcare professionals are more likely to indicate that workload/working hours (77%), unacceptable behavior of others, such as bullying and coercion (64%), and lack of support from colleagues or superiors to fulfill their role (61%), are causes of work-related stress [2].

The situation in healthcare can be harmful for healthcare professionals and their organizations [5,6,7]. In particular earlier research in the Netherlands [8] mentioned that the most burnout complaints occur in the occupational sectors education (22.1%) and healthcare (17.9%) and that these are caused by little autonomy, high workload, lack of support, and unacceptable behavior of others. Work-related stress has also a distressing economic impact: The Netherlands Organization for Applied Scientific Research (TNO) [9] estimated the costs of stress-related absenteeism for employers in the Netherlands to be €1.8 billion each year.

Amanullah, McNally, Zelin, Cole, and Cernovsky [10] concluded that hospital physicians who had significantly higher average scores on two subscales of the Maslach Burnout Inventory (emotional exhaustion and mental distance) benefited from prevention programs, such as those based on mindfulness or Cognitive Behavior Therapy. McCain, McKinley, Dempster, Campbell, and Kirk [11] confirmed the need for interventions to help doctors acquire appropriate coping mechanisms. They also reported high levels of burnout in primary and secondary care doctors, despite high levels of resilience. Yang, Meredith, and Kahn’s study [12] showed that higher levels of mindfulness among mental health professionals are associated with lower levels of stress and burnout (i.e., emotional exhaustion and mental distance).

Mindfulness-Based Stress Reduction (MBSR) [13], originally developed to relieve the suffering or stress of patients with chronic pain [14], is the most common form of secular mindfulness-based training [15]. It consists of eight 2.5-h weekly sessions and one 7-h day of silence. MBSR includes formal meditation exercises (the body scan, sitting meditation, walking and standing meditation, lying yoga exercises), and informal meditation exercises, paying full attention to daily activities. Mindfulness is defined as the awareness that arises through intentionally attending in an open, caring, and discerning way [16]. This definition integrates three elements. The first element is intention, referring to one’s personal goals and values, and reflecting “why” individuals pay attention. The second element is attention, i.e., attending to experiences in the here and now. The third element, attitude, relates to “how” individuals pay attention: in a non-judgmental way, with curiosity and compassion [17].

The literature demonstrates that mindfulness meditation has a positive impact on health and well-being in different populations, e.g., patients, healthy participants, students, and employees. Research initially reported positive benefits for various patient groups (e.g., those with chronic pain, anxiety, eating and major depressive disorders, fibromyalgia, psoriasis, or cancer) [18,19].

In a meta-analysis, Chiesa and Serretti [20] focused on the effects of MBSR on healthy participants and showed that mindfulness caused a significant reduction in stress levels. A more recent systematic review of the effects of MBSR on employees’ mental health [19] reported reduced levels of emotional exhaustion (a dimension of burnout), stress, psychological distress, depression, anxiety, and occupational stress. In addition, improvements were found for mindfulness, personal accomplishment (a dimension of burnout), (occupational) self-compassion, quality of sleep, and relaxation.

In research on the relationship between MBSR and employees’ mental health, the most commonly studied group of employees are healthcare professionals [19]. Six reviews [21,22,23,24,25,26] and three reviews/meta-analyses [27,28,29] focused exclusively on healthcare professionals and students. MBSR benefits the physical and mental health of different groups of healthcare professionals in various ways (e.g., decreasing stress levels, burnout and anxiety, increasing personal well-being and self-compassion, enhancing presence when relating to others, compassion, and a sense of shared humanity). In line with these findings, Lamothe, Rondeau, Malboeuf-Hurtubise, Duval, and Sultan [23] showed that MBSR decreases burnout, stress, anxiety, and depression, and improves empathy in healthcare professionals.

Another systematic review about mindfulness-based interventions in the workplace, performed by Lomas, Medina, Ivtzan, Rupprecht, and Eiroa-Orosa [24], examined the impact of mindfulness on the well-being of healthcare professionals. Although the results of some outcomes, such as burnout, are equivocal, overall, their review suggests that mindfulness improves the well-being of healthcare professionals: it decreases mental health issues (e.g., anxiety, burnout, depression, distress and anger, stress, and strain), increases well-being-related outcomes (e.g., compassion and empathy, emotional intelligence and regulation, health, mindfulness and awareness, relationships, resilience, well-being/satisfaction, and flourishing), and improves aspects of job performance. 

The objective of this study was to evaluate the feasibility and acceptability of the MBSR training and to explore both its positive and negative effects on symptom-focused mental health variables. In view of this, we studied a group of healthcare professionals (*n* = 30) at a hospital specialized in orthopedics, rheumatology, and rehabilitation. As such, we investigated the predictive validity of the MBSR training for this group of professionals.

With respect to the positive and negative symptom-focused mental health variables, two expectations were investigated. The first expectation was that MBSR training would increase positive symptom-focused mental health outcomes: total mindfulness, the five dimensions of mindfulness (observing, describing, acting with awareness, non-judging, and non-reactivity), personal accomplishment (a dimension of burnout), quality of sleep, positive emotions at work, self-efficacy, and work engagement. The second expectation was that the MBSR training would decrease negative symptom-focused mental health outcomes: emotional exhaustion and mental distance (two dimensions of burnout), stress, negative emotions at work, and worrying.

## 2. Methods

### 2.1. Study Design

This pilot study was designed as a non-randomized pre-post intervention. A questionnaire was administered one week prior to the start of the intervention (baseline: T_0_) and after the eight-week MBSR intervention period (post-intervention: T_1_). The study has been carried out in the Netherlands in accordance with the Declaration of Helsinki of 1975. All participants gave their informed consent for inclusion before they participated in this study.

As our psychological intervention approach has low risk for the participants, it was not required—in the period of the pilot study (2011–2012)—to send the protocol to a Research Ethics Committee. However in 2015 the PhD plan of the first author (M.J.), including the pilot study, was approved by the Ethical Committee of the HAN University of Applied Sciences (Registration no. ACPO 07.12/15). The randomized controlled trial—follow-up of this pilot study—is registered with the Dutch Trial Register (www.trialregister.nl): NL5581 (September 2016).

For feasibility reasons, we deviated from the original study protocol that included a control group as well, and conducted a quasi-randomized trial using a one-group pretest-posttest design [30].

### 2.2. Participants

Healthcare professionals (physicians, psychologists, physical therapists, nurses, social and pastoral workers, support workers, and managers) at a hospital specialized in musculoskeletal problems were approached in September, October, and November 2011 by the head of the Human Resources department and managers of other departments and invited to voluntarily participate in the pre-post intervention pilot study. The professionals willing to participate received a questionnaire that assessed a few inclusion criteria (i.e., being a healthcare professional in this specialized hospital; having worked there for at least two years, three days per week) and exclusion criteria (i.e., having attended mindfulness training over the past two years; having followed a stress reduction course, such as relaxation training or cognitive behavioral therapy, over the past two years). 

Thirty healthcare professionals participated in the intervention, divided among two training groups of 15 participants each. The MBSR training of the first group started in December 2011, and the training of the second group started eight weeks later (in February 2012).

### 2.3. Intervention

The MBSR training used is primarily based on the MBSR program developed by Jon Kabat-Zinn [13]. The program consists of eight 2.5-h weekly sessions and one 7-h day of silence during working hours. A very important part of the training is the homework: 45 min of daily practice at home, six days per week, with the support of guided CDs and a customized workbook. MBSR includes:guided instruction in mindfulness meditation practices (body scan, sitting meditation);simple movement exercises (stretching and yoga);a short group discussion;informal meditation exercises: paying full attention to daily activities (e.g., brushing one’s teeth, taking a shower, eating).

Two experienced mindfulness trainers (one being the first author) delivered the MBSR program together to the two groups. Both trainers meet the advanced criteria of the Center for Mindfulness of the University of Massachusetts Medical School (https://www.umassmed.edu/cfm/) and maintain regular personal meditation practices.

### 2.4. Outcome Measures

Our selection of outcome variables measuring mental health to be included in the study started from a list of the 15 most important mental health outcomes, as mentioned in the systematic review of Janssen, Heerkens, Kuijer, Van der Heijden, and Engels [19]. The level of evidence for the variables/outcomes was Level 2 (“it is plausible that …”), which implied that at least two medium-quality Randomized Controlled Trials (RCTs) show significance between groups differences, or Level 3 (“there are indications that …”), which meant that at least one medium-quality RCT shows significance. In the systematic review (SR), we found no studies that met the requirements for Level 1 (“it has been proven that …”), which referred to significance between groups in at least two high-quality RCTs [19].

The selection of the mental health variables was as follows. First, we chose the three most important variables: mindfulness, burnout, and stress level. Second, except for one additional negative symptom-focused variable (negative emotions), we selected only positive symptom-focused variables: quality of sleep, positive emotions, self-efficacy, and work engagement. The main reason is that, as reported in Janssen et al.’s SR [19], most MBSR studies use negative symptom-focused variables, although positive outcomes may also indicate or contribute to well-being. Finally, one variable for which there was no evidence in the SR (Level 5)—worrying—was chosen as well. There are two reasons for this choice: (1) there were indications that many participants were worrying; and (2) Mindfulness-Based Cognitive Therapy, the most important adaptation of MBSR [31], decreases worrying [32].

### 2.5. Measurement Instruments

Information on the feasibility and acceptability of the training were collected by assessing the recruitment rate, the retention rate, and the participants’ evaluation of the training by means of five multiple choice items, followed by some questions in open-ended format [33]. The following open questions were used to gather additional information: “Did the training meet your expectations? Does the training have a positive effect on your daily life? Are you satisfied with the content and structure of the training? Are you satisfied with the educational methods of the training? Are you satisfied with the trainer(s)?”

The following well-validated questionnaires (in Dutch) were employed:the Dutch version of the Five Facet Mindfulness Questionnaire (FFMQ-NL) [34];the Dutch version of the Maslach Burnout Inventory—General Survey (MBI-GS): the Utrechtse BurnOut Schaal—Algemeen (UBOS-A; Utrecht Burnout Scale—General) [35,36,37];the stress scale of the Dutch Depression, Anxiety, and Stress Scales (DASS) [38];the Dutch sleep quality subscale of the Vragenlijst Beleving en Beoordeling van de Arbeid (VBBA; Questionnaire Perception and Assessment of Labor) [39];the Dutch version of the Job-related Affective Well-Being Scale (JAWS) [40,41];the Dutch General Self-Efficacy Scale, a translated version of the original German instrument [42,43,44,45];the Dutch version of the shortened Utrecht Work Engagement Scale (UWES), the UBES-9 [46,47];the Dutch VBBA worrying subscale [39].

### 2.6. Instrument Selected to Measure the Primary Outcome Measure

*Mindfulness skills* were examined with the FFMQ-NL [18,34]. The 39-item FFMQ-NL has a five-factor structure, which is captured in the following five subscales: *observing*, *describing*, *acting with awareness*, *non-judging* of inner experience, and *non-reactivity* to inner experience. The FFMQ-NL total score is ranging from 39 to 195; the total scores of the subscales, except *non-reactivity* (7 to 35), are 8 to 40. Higher values indicate more mindfulness skills. The internal consistency (Cronbach’s alpha) for the FFMQ-NL total score is 0.85 (for the non-meditating sample) and 0.90 (for the meditating sample); the Cronbach’s alphas for the five subscales vary from 0.70 to 0.89. The five dimensions show modest but significant correlations among one another (ranging from 0.13 to 0.39), which suggests that they represent distinct but interrelated constructs. Overall, the psychometric properties of the FFMQ-NL [34] are comparable to the original English version [18].

### 2.7. Instruments Selected to Measure the Secondary Outcome Measures

*Burnout* was measured using the Dutch version of the MBI-GS: the UBOS-A; Utrecht Burnout Scale—General [35,36,37]. The 15-item UBOS-A has a three-dimensional structure with three subscales: *emotional exhaustion*, *mental distance* (cynicism, depersonalization), and (job-related) *personal accomplishment*.

The total scores of the three subscales are ranging from 0 to 6. Higher values indicate more emotional exhaustion, more mental distance, and more personal accomplishment, respectively. Cronbach’s alphas of the three subscales—emotional exhaustion (5 items), mental distance (4 items), and professional efficacy (6 items)—are 0.88, 0.81, and 0.75, respectively.

*Stress* was assessed with the 14-item stress scale of the Dutch 42-item Depression, Anxiety, Stress Scales (DASS). The total score on the stress scale is ranging from 0 to 21. Higher values indicate more stress. The DASS has a three-factor structure: depression, anxiety, and stress. The internal consistency (Cronbach’s alpha) of the DASS factors is 0.94, 0.88, and 0.93, respectively [38]. 

*Quality of sleep* was measured using the Dutch sleep quality subscale of the 14-item VBBA (Questionnaire Perception and Assessment of Labor). The total score is ranging from 0 to 100. Higher values indicate less quality of sleep. The internal consistency (Cronbach’s alpha) is 0.86 [39].

*Positive and negative emotions at work* were assessed by the 12-item Dutch version of the JAWS [40,41]. The Dutch JAWS has a two-factor structure, which led to the following two subscales: a positive six-item emotions scale (Cronbach’s alpha = 0.77) and a negative six-item emotions scale (Cronbach’s alpha = 0.78). The total score on each subscale is ranging from 6 to 30. Higher values indicate more positive emotions and more negative emotions, respectively [41].

*Self-efficacy* was assessed using the Dutch General Self-Efficacy Scale, a translated 10-item version of the original German instrument [42]. The total score is ranging from 10 to 40. Higher values indicate more self-efficacy. The German scale has an internal consistency ranging from 0.75 to 0.91 [43,44]. The Cronbach’s alpha coefficient of the Dutch version is 0.85 [45].

*Work engagement* was assessed using the nine-item Dutch version of the shortened Utrecht Work Engagement Scale (UWES), the UBES-9 [46,47]. The three-dimensional UWES consists of three 3-item subscales: vigor, dedication, and absorption. The total score of the UWES is ranging from 9 to 45. Higher values indicate more work-engagement. The internal consistency (Cronbach’s alpha) for the total UBES-9 is 0.93 and the alphas for the three subscales vary from 0.79 to 0.89. The three work engagement scales are highly correlated (minimum = 0.65). The three factors are negatively correlated with the three dimensions of burnout [46]. 

*Worrying* was measured using the Dutch four-item VBBA worrying subscale. The total score is ranging from 0 to 100. Higher values indicate more worrying. The internal consistency (Cronbach’s alpha) is 0.80 [39].

### 2.8. Statistical Analyses

Normality of data was checked and verified by histograms, normal probability plots, and Shapiro–Wilk tests [48]. To examine the effects of MBSR training (differences between baseline and post-intervention) on mental health, we used *t*-tests for paired samples. Given the non-normally distributed variables on baseline and post-intervention, we also used the Wilcoxon signed-rank test. In this case, significance on the Wilcoxon signed-rank test was a requirement to accept significance on the *t*-test for paired samples (Table 1). Two-tailed tests were performed. 

Effect sizes for the difference between means are reported: small (*d* = 0.20), medium (*d* = 0.50), and large (*d* = 0.80) [49,50].

All statistical analyses were performed using IBM SPSS Statistics (Version 23). The level of significance was set at 0.05.

## 3. Results

The research population, a heterogeneous sample drawn from various occupational categories consisted of seven psychologists, two physical therapists, two nurses, five social/pastoral workers, five support workers, and nine managers. The mean age of the participants, consisting of 6 males and 24 females, was 44.5 years (range 27 to 64 years).

Regarding the recruitment phase, we received more applications (exceeding 40) than we offered places (30). Hence, a numerus fixus was needed; only the first 30 applications were admitted.

Of the 30 professionals, 29 (6 males and 23 females) completed the survey measure at both time points. The participation (“retention rate”) of the 29 participants in the intervention study sessions can be summarized as follows: 17 of them participated in all nine sessions, 10 in eight sessions, 1 in seven sessions, and 1 in six sessions (*M* = 8.50, *SD* = 0.74). The 30th participant, a female manager, attended two sessions. 

In Table 2 the participants’ evaluation of the MBSR training is included. The participants’ evaluation of the training (first question) showed that for 83% of the participants the training has met their expectations (“reasonable degree” and “absolutely”). For 62% of the participants the training had a positive effect on their daily life (second question). For the last three questions these percentages were 90%, 90%, and 97%, respectively. 

The mental health variables in Table 1 are presented in order of importance [19]. Two criteria are considered: first, the level of evidence, and second, the number of studies reporting a particular (significant or non-significant) result [19]. As shown in Table 1, statistically significant improvements were observed on many outcome variables, from baseline to post-intervention. Significant increases in the total mindfulness score and in four mindfulness dimensions, except “describing”, were reported. The effect sizes were medium to large. Significant increases with medium effect sizes were also found for personal accomplishment, quality of sleep, positive emotions, and self-efficacy. Significant decreases, at least medium, were reported for stress level, negative emotions, and worrying. No significant improvements were observed for the variables: describing, emotional exhaustion, mental distance, and work engagement.

## 4. Discussion

The purpose of this pilot study was to measure the feasibility and acceptability of the current trial and to examine both positive and negative mental health variables which could be used to test the predictive validity of a MBSR training in a group of healthcare professionals. Two clustered expectations were investigated. The first expectation was that MBSR training would increase positive symptom-focused mental health outcomes. The second expectation was that MBSR training would decrease negative symptom-focused mental health outcomes.

Of the 130 contacted healthcare professionals, 40 showed interest for 30 available places. Of the 30 participants, 29 completed the survey measure at both time points. The degree of participation during the eight week MBSR program suggests that the participants showed willingness as reflected in an average total participation time of 8.5 sessions out of the nine sessions that were offered. Training compliance with the 45 min of daily practice at home and performing informal meditation exercises, during and after the training, was not assessed or reported. The participants’ evaluation of the training was positive with regard to the content and the course of the training, but only moderately positive with respect to the influence of the training on daily life. A limitation is that the items to evaluate the MBSR training had only four scale points using rather rough wordings (“Not at all”, “Somewhat”, “Reasonable degree”, “Absolutely”; see Table 2), which have prevented a more precise assessment.

Between baseline and post-intervention, the participants, a group of healthcare professionals (*n* = 30) at a specialized hospital, showed a significant improvement in general mindfulness (specifically, in four out of five mindfulness dimensions (except describing)), personal accomplishment, quality of sleep, positive emotions at work, and self-efficacy, and a significant decrease in emotional exhaustion (dimension of burnout), stress level, negative emotions at work, and worrying. These results, except for worrying, are in accordance with the findings in published reviews/meta-analyses [19,20,21,22,23,24,25,26,27,28,29]. There is plausible evidence for the effectiveness of the intervention for general mindfulness, emotional exhaustion and personal accomplishment (two dimensions of burnout), stress level, and quality of sleep, and indicative evidence for its effectiveness in the light of positive and negative emotions at work [19]. The significant decrease in worrying is remarkable, because no previous reviews/meta-analyses [19,20,21,22,23,24,25,26,27,28,29] have mentioned this. Thus far, only studies on Mindfulness-Based Cognitive Therapy (MBCT) have reported decreases in worrying in relation to mindfulness training [31]. A possible explanation is that one of the MBSR trainers, who is educated in MBCT, introduced the CT component of MBCT in response to the worrying complaints of many participants. The CT component includes psychoeducation about the nature of thoughts (“thoughts are not facts”) and the explicit link between one’s thoughts and one’s mood. It is possible that the decrease in worrying might have been caused by this additional CT component. However, we found no significant changes for mental distance (dimension of burnout) and work engagement. In addition, the reliability of the instruments to measure mental health variables appeared adequate (see Cronbach’s alpha in Table 1).

All mental health variables used (mindfulness, burnout, stress level, quality of sleep, positive emotions at work, negative emotions at work, self-efficacy, and worrying), except for work engagement, showed statistically significant effects of the MBSR training and seemed appropriate for a larger-scale RCT on its effects. When one dimension of a variable showed statistically significant effects, for sake of incorporating a broad scope of possible explaining variables, we argue that the variable should be included in follow-up research. Moreover, although work engagement revealed no statistically significant effects, we posit that the variable will still be appropriate for a larger-scale RCT as it comprises a positive counterpart of burnout [51]. We assume that in the daily healthcare context the sharing of positive experiences regarding stress reduction and burnout prevention at the workplace are just as important as statistically significant outcomes. For this reason, we not only reported the *p*-values in Table 1, but also the effect sizes [52]. For interested readers, we refer to Copay et al. [53] who discuss the concept of “minimal clinically important difference” (MCID) and go into methods to assess this. So far, there is no academic consensus about the most appropriate method.

The participants of this study have different professions and different job characteristics, e.g., various weekly work hours, work shift/no work shift, patient/no patient contact, supervisor/non-supervisor. That may influence mental health variables’ ratings at baseline (e.g., emotional exhaustion, and stress level) and the opportunity to participate in the MBSR training. For this study, these job characteristics were not available and also some demographic data are lacking (e.g., education level, profession, years of experience, area of work, and previous experience with MBSR or meditation). This may be considered as a limitation of this pilot study. In a larger-scale RCT, such demographics will give researchers the opportunity to provide a more in-depth answer to the question of “which participants can benefit most from MBSR training?” In addition, it would be useful to examine the effects of MBSR training on experiencing job characteristics.

Training compliance as regards the 45 min of daily practice at home and performing informal meditation exercises was not reported. In a systematic review and meta-analysis, Parsons, Crane, Parsons, Fjorback, and Kuyken [54] reported a significant but small positive correlation between home practice and intervention outcomes. This should be considered in a larger-scale RCT too.

In addition, we only measured short-term effects, immediately after the MBSR training. Attending a MBSR training course requires a real investment (e.g., time, energy, and financial costs) and, from this perspective, examining long-term effects has high priority. We used questionnaires to measure variables and collect quantitative data using self-ratings. The reliance on these self-report data, however, may be subject to response bias. This study contained no qualitative data, which reduces our ability to investigate in-depth the participants’ experiences with the training and daily practice at home. There were no symptom-focused outcome measures related to work performance (e.g., caring efficacy, work behavior, work performance, and work ability). A few variables (mindfulness, self-efficacy, and worrying) were assessed using process-focused (mediating) measures, which may be suitable for capturing the mechanisms by which formal and informal mindfulness practice leads to specific outcomes, like mental and physical health and work performance. Moreover, no outcomes referred to perceptions of work characteristics (e.g., work pressure, emotional load, feedback, autonomy, and learning opportunities). That should be considered in a larger-scale RCT as well.

MBSR is a person-centered intervention, which seems to be only partly effective at improving employees’ mental health [55]. However, stress can be caused both by personal characteristics and work characteristics. Therefore, an integrated approach that also considers the work context is needed [56].

To address most of these limitations, a larger-scale RCT [57] should assess the short- and long-term effects on a diversity of outcomes: negative and positive symptom-focused measures of mental health; work performance; process-focused outcome measures, and outcomes on work-related perceptions. Moreover, a larger, homogeneous sample or stratification is needed to be able to detect specific, statistically significant, demands and challenges of these groups with respect to the outcomes of MBSR training. Qualitative data are also needed to profoundly investigate the participants’ experiences. The moderating effect of an additional organizational intervention, which may enhance the effects of the MBSR training, should be examined.

## 5. Conclusions

Significant improvements were found in general mindfulness, personal accomplishment, quality of sleep, positive emotions, and self-efficacy. A significant decrease was found in emotional exhaustion, stress level, negative emotions at work, and worrying. The results of this pilot study suggest that MBSR may help to improve employees’ mental health. However, no firm conclusions can be drawn.

The trial that is dealt with in this study seems to be feasible and acceptable and forms a sound basis for implementation in follow-up empirical research. The results suggest that mindfulness, burnout, stress level, quality of sleep, positive emotions at work, negative emotions at work, self-efficacy, and worrying are meaningful mental health variables, which makes it appropriate for a larger-scale RCT intended to investigate the effects of MBSR.

## Figures and Tables

**Table 1 ijerph-17-09420-t001:** Mental health variables’ ratings before and after the Mindfulness-Based Stress Reduction (MBSR) training.

Mental Health Variables	*α* T_0_	*α* T_1_	Min−Max T_0_ Min−Max T_1_	*n*	T_0_: *M (SD)*	Normality^+^ T_0_	T_1_: *M (SD)*	Normality^+^ T_1_	Wilcoxon Z (*p* 2-Tailed)	Difference T_1_–T_0_ *M (SD)*	*ft (df)*	*p* *t*-Test (2-Tailed)	Effect Size
FFMQ−NL	0.91	0.92	88.92–159.12 102.96−171.99	28	125.61 (17.09)	0.98 (0.94)	137.14 (15.92)	0.98 (0.83)		11.53 (14.44)	164.78 (27)	**<0.01**	0.80
Observing	0.80	0.70	14.00–34.00 21.04–34.00	28	25.86 (5.21)	0.95 (0.24)	28.10 (3.39)	0.97 (0.60)		2.25 (4.16)	22.90 (27)	**<0.01**	0.54
Describing	0.81	0.91	17.04–35.04 14.00–37.04	28	26.54 (4.17)	0.99 (0.96)	27.50 (5.43)	0.93 (0.10)		0.96 (4.28)	9.54 (27)	0.24	0.23
Acting awareness	0.88	0.91	14.00–38.00 18.00–39.04	28	24.04 (5.34)	0.97 (0.68)	26.00 (5.43)	0.95 (0.28)		1.96 (4.26)	19.50 (27)	**0.02**	0.46
Non−judging	0.89	0.93	16.00–38.00 20.00–40.00	28	27.82 (6.14)	0.97 (0.56)	31.25 (5.64)	0.96 (0.43)		3.43 (4.02)	36.08 (27)	**<0.01**	0.85
Non−reactivity	0.78	0.80	14.98–30.03 17.01–31.99	28	21.36 (4.09)	0.96 (0.37)	24.29 (3.62)	0.98 (0.75)		2.93 (3.96)	27.38 (27)	**<0.01**	0.74
UBOS, emotional exhaustion subscale	0.84	0.89	0.60–4.40 0.00–4.20	27	1.99 (1.08)	0.92 (0.05)	1.64 (1.00)	0.92 (0.05)	−1.84 ^a^ (0.07)	−0.34 (1.09)	−1.63 (26)	0.012	0.31
UBOS, personal accomplishment subscale	0.82	0.85	1.50–5.67 2.17–6.00	27	4.00 (0.88)	0.96 (0.36)	4.28 (0.94)	0.94 (0.14)		0.28 (0.50)	2.94 (26)	**<0.01**	0.57
UBOS, mental distance subscale	0.84	0.86	0.25–4.75 0.00–4.50	27	1.90 (0.96)	0.97 (0.62)	1.62 (1.06)	0.92 (0.04)	−1.62 ^a^ (0.11)	−0.28 (0.89)	−1.61 (26)	0.12	0.31
DASS, stress subscale	0.91	0.86	0.49–12.53 0.00–9.52	28	4.79 (3.28)	0.91 (0.03)	3.21 (2.34)	0.92 (0.04)	−2.52 ^a^ **(0.01)**	−1.57 (3.07)	−18.94 (27)	**0.01 ^1^**	0.51
VBBA, sleep quality subscale (complaints)	0.79	0.77	0–71 0–71	27	27.25 (22.06)	0.92 (0.04)	19.05 (19.00)	0.86 (0.00)	−2.68 ^a^ **(<0.01)**	−8.20 (13.97)	−305.10 (26)	**<0.01 ^1^**	0.59
JAWS, positive emotions subscale	0.77	0.94	12.00–27.00 9.00–30.00	28	21.68 (3.35)	0.94 (0.13)	23.36 (4.62)	0.94 (0.12)		1.68 (3.59)	14.84 (27)	**0.02**	0.47
JAWS, negative emotions subscale	0.75	0.75	7.98–28.02 9.00–24.00	28	15.86 (4.08)	0.86 (0.00)	13.07 (3.09)	0.92 (0.04)	−3.30 ^a^ **(<0.01)**	−2.79 (4.05)	−21.84 (27)	**<0.01 ^1^**	0.69
Dutch General Self−Efficacy Scale	0.86	0.90	24–40 27–40	28	31.43 (4.23)	0.98 (0.86)	33.04 (4.01)	0.95 (0.20)		1.61 (3.46)	24.60 (27)	**0.02**	0.47
UWES	0.86	0.86	18.99–42.03 17.01–42.03	27	30.22 (5.65)	0.98 (0.89)	30.63 (5.41)	0.95 (0.22)		0.41 (3.90)	4.89 (26)	0.60	0.11
VBBA, worrying subscale	0.71	0.72	0–100 0−100	27	37.96 (34.23)	0.87 (0.00)	24.07 (30.60)	0.77 (0.00)	−2.52 ^a^ **(0.01)**	−13.89 (24.35)	−296.40 (26)	**<0.01 ^1^**	0.57

*α* Cronbach’s alpha. Normality^+^: statistic Shapiro–Wilk test (*p* value); *p* value > 0.05 indicates a normal distribution. ^a^ Wilcoxon signed-rank test: based on positive ranks. ^1^. Wilcoxon signed-rank test and *t*-test showed significance. Normality of the data was checked and verified by histograms, normal probability plots, and Shapiro–Wilk tests. To examine the effects of MBSR training on the incorporated mental health outcomes, we used *t*-test for paired samples, including effect sizes, for normally distributed mental health outcomes. The Wilcoxon signed-rank test and *t*-test for paired samples were used for non-normally distributed mental health outcomes. In this case, the outcomes of the *t*-test for paired samples can only be used if the Wilcoxon signed-rank test showed significance. Bold: significant (two-tailed).

**Table 2 ijerph-17-09420-t002:** The participants’ evaluation of the MBSR training.

Items	Not at All	Somewhat	Reasonable Degree	Absolutely
Did the training meet your expectations?	0	5	16	8
Does the training have a positive effect on your daily life?	0	11	13	5
Are you satisfied with the content and structure of the training?	0	3	16	10
Are you satisfied with the educational methods of the training?	0	3	17	9
Are you satisfied with the trainer(s)?	0	1	18	10

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
