# Peer review of "Effects of Mindfulness-Based Stress Reduction Training on Healthcare Professionals’ Mental Health: Results from a Pilot Study Testing Its Predictive Validity in a Specialized Hospital Setting"

_ijerph, 2020, doi:10.3390/ijerph17249420_

Round 1
Reviewer 1 Report
The manuscript presented is very interesting. And it proposes research that can help improve the quality of life of workers.
The work is methodologically very well done.
However, it is surprising that the authors have decided to publish now a paper with data from 2011.
However, there are some aspects of the text that could be improved to increase its understanding and usefulness:
Introduction:
-Page 3, l.69: For better understanding it would be relevant to increase the information about the MBSR system.
-P.5, l.114: Describe the objective of the research conducted.
Discussion:
Reflect on how the fact that professionals have different health professions might influence.
Reflect on how incorporating an additional technique into one of the groups might compromise the results of the study.
Conclusions:
- Expand the conclusions for a better understanding of the work presented. Not only justify the need for an RCT. The work seems to be a preliminary one that justifies the author's doctoral thesis, but it must have entity by itself.
Author Response
Response to reviewer 1
Dear Reviewer, thank you for your insightful feedback and comments that have helped us to further improve the manuscript. Below, we will go into the point-by-point reply to all issues raised.
-Page 3, l.69: For better understanding it would be relevant to increase the information about the MBSR system.
Thank you for this advice. We have changed the text (Line 72 to 74). In Line 149 – 163 you will find an extended description of the MBSR training.
-P.5, l.114: Describe the objective of the research conducted.
In line 106 and 108 the purpose of the study if given. We have changed the term purpose in objective.
We have shortened the first sentence of paragraph 2.1 as the objective is already given in line 106-108.
Reflect on how the fact that professionals have different health professions might influence.
If we understand you correctly, you want to know whether the profession of the participants has influenced the results. That is an interesting question, which we have tried to answer in Line 349 to 355.
Reflect on how incorporating an additional technique into one of the groups might compromise the results of the study.
Thank you for this observation. You are right and for this reason we have changed the text both in paragraph 2.4 (Line 160 to 161) as well as in paragraph 2.4 Line 327 to 332.
Conclusions:
- Expand the conclusions for a better understanding of the work presented. Not only justify the need for an RCT. The work seems to be a preliminary one that justifies the author's doctoral thesis, but it must have entity by itself.
Thank you for this advice. We have added two sentences about the pilot (Line 391 to 395).

Reviewer 2 Report
Mindfulness based stress reduction programs have been demonstrated to have a significant impact on the well-being of various groups. The current pilot study was designed to test the efficacy and acceptance with a group of health care providers. Clearly this is a timely study and a more advance RCT with health care workers is warranted.
I am, for the most part, in favor of publication of your pilot study. There are, however, some issues to be addressed before the manuscript is ready for publication. Below I outline my concerns beginning with the greatest of the issues I feel need to be addressed.
Table 1 presents the participants' evaluation of the MBSR training. In the text you wrote that 85% of the evaluations of the training were positive. This appears to be so regarding satisfaction with the training, but 38% of the participants reported that the training only had "somewhat" of positive effect on their daily life. I believe this needs to be addressed as this is not a very 'positive' outcome. Also, the psychometric properties of the four response alternatives are questionable. I feel this should be addressed.
Along similar lines, although this is a pilot study and the sample size is quite small, it is still important to look at the reliability of the questionnaires used in your study in the current sample.
You state that 40 applications were submitted for the program to fill the 30 available positions. However, you do not disclose how many potential participants were recruited. There is no way to judge whether the program seemed acceptable to the health care workers at the institution without knowing the proportion of potential participants that applied for the program.
Some minor issues/questions:
In line 51 you state that health care workers are more likely to experience "mobbing" compared to other categories of workers. What is mobbing?
On the same page in line 56 you wrote that burnout is caused by unacceptable behaviors. I assume that you are referring to the unacceptable behaviors of others.
Overall I found this to be sound pilot study that does support the need for a larger scale RCT study. I hope that my suggestion above help to improve the current manuscript.
Author Response
Response to reviewer 2
Dear Reviewer, thank you for your insightful feedback and comments that have helped us to further improve the manuscript. Below, we will go into the point-by-point reply to all issues raised.
Table 1 presents the participants' evaluation of the MBSR training. In the text you wrote that 85% of the evaluations of the training were positive. This appears to be so regarding satisfaction with the training, but 38% of the participants reported that the training only had "somewhat" of positive effect on their daily life. I believe this needs to be addressed as this is not a very 'positive' outcome. Also, the psychometric properties of the four response alternatives are questionable. I feel this should be addressed.
Thank you for this remark. You are right and we have reworded the text in the abstract (Line 25 to 26), in the results (Line 273 to 277) and in the discussion (Line 310 to 315).
Along similar lines, although this is a pilot study and the sample size is quite small, it is still important to look at the reliability of the questionnaires used in your study in the current sample
Thank you for this advice. We think that the reliability of the instruments to measure mental health variables is adequate; see Cronbach’s alpha in table 2 on T0 and T1.We have changed the text (Line 333 to 335).
You state that 40 applications were submitted for the program to fill the 30 available positions. However, you do not disclose how many potential participants were recruited. There is no way to judge whether the program seemed acceptable to the health care workers at the institution without knowing the proportion of potential participants that applied for the program.
Thank you for this remark and of course you are right. The hospital is quite large but we were enthusiastic about the eagerness to participate. We have removed the sentence about our ‘successfulness’.
In line 51 you state that health care workers are more likely to experience "mobbing" compared to other categories of workers. What is mobbing?
Mobbing is a synonym for bullying; as this latter term is probably more common, we have now used the term bullying (Line 52).
On the same page in line 56 you wrote that burnout is caused by unacceptable behaviors. I assume that you are referring to the unacceptable behaviors of others.
Thank you for this advice. We have added ‘of others’ after unacceptable behavior(s) (Line 52 & Line 57).
